# MicroRNAs as Biomarkers in Canine Osteosarcoma: A New Future?

**DOI:** 10.3390/vetsci7040146

**Published:** 2020-09-30

**Authors:** Olivia Gourbault, Lola Llobat

**Affiliations:** 1Facultad de Veterinaria, Universidad Cardenal Herrera-CEU, CEU Universities, 46113 Valencia, Spain; oligou@alumnos.uchceu.es; 2Research Group Microbiological Agents Associated with Animal Reproduction (PROVAGINBIO), Department of Animal Production and Health, Veterinary Public Health and Food Science and Technology (PASAPTA) Facultad de Veterinaria, Universidad Cardenal Herrera-CEU, CEU Universities, 46113 Valencia, Spain

**Keywords:** biomarkers, canine sarcoma, canine osteosarcoma, canine tumour, dog, veterinary oncology

## Abstract

Sarcomas are frequent in dogs and canine species are excellent animal models for studying the human counterpart. However, osteosarcomas are a rare form of sarcoma with high death rates in humans and dogs. miRNAs are small endogenous RNAs that regulate gene expression post-transcriptionally. The discovery of miRNAs could give a contribute in the diagnosis and prognosis of different types of tumors in animal species, as already in humans. The differentiated expression of miRNAs is a frequent finding in cancers and is related to their pathogenesis in many cases. Most canine and human sarcomas show similar miRNA aberrations. Lower levels of miR-1 and miR-133b in canine osteosarcoma tissues were found to increase tumorigenesis through a higher expression of their target genes MET and MCL1. The overexpression of miR-9 promotes a metastatic phenotype in canine osteosarcomas and its capacity as a prognostic biomarker for the disease is currently being evaluated. MicroRNAs at the 14q32 locus could be used as prognostic biomarkers, since their decreased expression has been associated with poor prognosis in canine and human osteosarcomas. Furthermore, a decreased expression of miR-34a in osteosarcoma tumour cells has been associated with shorter disease-free survival times and its reintroduction as a synthetic prodrug shows good potential as a novel therapeutic target to fight the disease. Circulating miR-214 and miR-126 are significantly increased in a broad-spectrum cancer and have the ability to successfully predict the prognosis of dogs. However, further studies are needed to make the use of miRNAs as biomarkers a common practice.

## 1. Introduction

Sarcomas are a rare type of cancer that arise from mesenchymal stem cells [1]. In dogs, sarcomas encompass 10%–15% of malignant tumours and have a rapid progression [2,3]. Canine sarcomas are excellent models for comparative oncology as they arise spontaneously within an intact immune system, are exposed to similar environmental factors, and their anatomy and physiology are more closely related to humans than other more traditionally used animal models such as rodents [2,4,5,6,7].

## 2. Canine Osteosarcoma

Osteosarcomas (OSA) are the most common bone tumour in dogs and humans, comprising the fifth most common cause of cancer in adolescents aged 15–19 [8,9]. Compared with other species OSA occurs more frequently in dog, representing up to 90% of primary bone tumours and usually affects dogs older than 6 years [10]. The incidence in dogs is approximately 139 per 100,000, and is higher in large (25–45 Kg) and giant (>45 Kg) breeds with a particularly high predisposition in Rottweilers, Greyhounds and Irish Wolfhounds [6,10,11,12,13]. OSA in dogs and humans share many similar clinical and molecular characteristics. For example, in both species, OSA predominantly occurs in the appendicular skeleton (up to 80% in dogs and 90% in humans) than in the axial skeleton [2,6,11,14,15]. OSA in humans can be compared to that of dogs with respect to histological findings, its high rate of metastasis, and its low long-term survival rates [16]. Likewise, it has been seen that OSA, in both species, has similar transcription profiles and the same aberrations in the number of copies of DNA (40). In fact, canine OSA present abnormal karyotypes and DNA copy number aberrations [17,18,19]. Therefore, the exploration of new biomarkers in dogs with OSA could also be beneficial for human patients [20]. However, there are also differences between the two species, one of the main clinical differences being the age of onset of the disease, usually appearing in adolescence in humans, while in dogs, they tend to occur more frequently in adulthood (mean age at diagnosis of approximately eight years) [10]. The treatment in both species consists of an aggressive chemotherapy and the surgical excision of the tumour, which, in dogs, entails the amputation of the affected limb [6,10]. Osteosarcomas most commonly metastasise to the lungs and is the main cause of death [2,5,6,10]. In humans, the presence of metastasis at diagnosis has a significant impact on prognosis, with a 5-year survival rate than decreases from 70% with the localized disease to 20% [6,10,14]. The prognosis in dogs is worse, with a 1-year survival rate below 45% in patients with localized disease, dropping to a survival time of only 76 days in dogs with metastasis at the time of diagnosis [6].

## 3. What Are MicroRNAs?

Recently, a new group of molecules emerged as genetic biomarkers, microRNAs (miRNAs). The first miRNA (lin-4) was discovered in 1993 in the nematode *Caenohabditis elegans* but its importance was only understood after the discovery of a second miRNA, let-7 [21,22]. MiRNAs are small, non-coding RNA molecules of approximately 20–24 base pairs in length, are therefore they are unable to code for proteins [23,24,25]. They are usually phylogenetically conserved and have been shown to play a crucial role in the regulation of gene expression and cellular processes [25,26]. Since their discovery, thousands of miRNAs have been identified and carrying out different functions in different organisms. Studies show that their presence or absence in different cancers can serve as potential biomarkers for the disease. In addition, different miRNAs also have shown their potential as new therapeutic cancer targets [25].

### 3.1. Biogenesis of MiRNAs

The biogenesis of miRNAs is a complex process that requires two endonucleases [27]. It begins in the cell nucleus, where specific genes give rise to miRNA that are transcribed by RNA polymerase II or III as primary transcripts (pri-miRNA). The enzyme Drosha and its cofactor DGCR8/Pasha then cut the pri-miRNA molecule to form a pre-miRNA, a structure comprising of 70–90 nucleotides in the form of a stem-loop [25,28,29,30,31]. This pre-miRNA is actively transported from the nucleus to the cytoplasm by the nuclear transport receptor Exportin 5 (XPO5) in a Ran-GTP protein dependent manner by XPO5 [32,33,34]. Exportin 5 overexpression has been reported to improve transport efficiency and enhance mature miRNA expression [34]. XPO5 plays an important role in the development of various cancers, such as hepatocellular carcinoma, colorectal cancer and non-small cell lung cancer [35]. Once in the cytoplasm, the pre-miRNA is cut by a second enzyme called Dicer (regulated by XPO5), until the formation of a mature and short double-stranded miRNA molecule (approximately 22 nucleotides long) [36,37,38,39,40,41,42]. The miRNA duplex unwinds, and is incorporated into the RISC protein complex, which is responsible for directing the silencing of messenger RNA and leads to the repression of gene expression [25]. This regulation appears to be post-transcriptional, via the inhibition of protein translation or the destabilization of target transcripts [41]. Figure 1 shows a diagram of miRNA biogenesis.

### 3.2. MiRNAs as Biomarkers

MiRNAs are clustered in families, whose members are evolutionarily related. Around 296 miRNAs belonging to 177 conserved families among placental mammals are currently known [43]. In fact, most miRNAs that appear to be poorly conserved are actually poorly understood and tend to derive from the duplication and divergence of an ancestral gene or the de novo appearance of a new gene [44,45]. Moreover, most of the targets of these miRNAs are highly conserved also in mammals, which facilitates the comparison of their study between different species [46].

MiRNAs regulate a large number of transcription factors and have great significance in the control and regulation of cell growth and development, proliferation and apoptosis [25,47]. In fact, their aberrant expressions have been repeatedly seen in various diseases, including different types of cancer [25]. Some studies indicate that miRNAs may play a causal role in tumour formation, functioning as tumour suppressors or oncogenes by targeting genes involved in tumour development, progression, or in the formation of metastasis [11].

## 4. MiRNAs in Canine Osteosarcoma Cells

Studies on gene expression profiles have demonstrated alterations in the expression of miRNAs in a wide range of human diseases [48]. OSA in humans and dogs have significant clinical, histological, and molecular similarities, such as its predilection for appendicular bones, its metastatic phenotype (towards the lungs and bones), the presence of indistinguishable lung micro-metastasis at the time of diagnosis, its multimodal therapy, and the regulatory change of several critical molecular pathways such as the p53, PTEN, and MYC pathways, among others. On the other hand, genetic analysis of OSA in humans and dogs demonstrates a high degree of reorganization with conserved genomic alterations between species and strikingly similar gene expression profiles [49]. There is considerable evidence of the aberrant expression of miRNAs in cancers compared to their expression in normal cells. By targeting genes involved in different cancerous processes, miRNAs have the ability to act as both oncogenes or tumour suppressor genes. They are capable of avoiding the action of growth suppressor genes, resisting cell death, activating an aggressive phenotype and inducing angiogenesis [16,49,50,51]. These characteristics make miRNAs potential candidates as novel biomarkers and therapeutic targets in the fight against osteosarcoma [16] (Table 1). 

### 4.1. miR-1 and miR-133b

Studies have shown that osteosarcoma tissues in humans and dogs show a reduced expression of miRNAs miR-1 and miR-133b in comparison to normal bone tissue. This is associated with a higher expression of their target genes MET and MCL1 [16,52,53]. The overexpression of MET, a target gene for both miRNAs, is present in 70%–80% of human osteosarcomas and contributes to an aggressive phenotype. In both canine and human tumour cells, overexpression of MET causes the activation of different signalling cascades, resulting in the growth, proliferation, invasion and survival of tumour cells. MCL1, a target gene for miR-133b, is a protein with anti-apoptotic properties. It is expressed in a wide variety of human tissues and neoplastic cells, and plays an important role in the development of various malignant tumours [16]. These results highlight the potential of miR-1 and miR-133b as candidates to act as potential diagnostic biomarkers in canine osteosarcoma.

### 4.2. miR-9

Another miRNA, whose overexpression seems to contribute to the aggressive biological behaviour of osteosarcomas is miR-9. The proteomic analysis carried out by Fenger et al. (2014) identified several miR-9 regulated proteins, such as, the actin filament cutter protein gelsolin [11]. Gelsolin is capable of inducing a gene expression pattern, that leads to increased cell motility and invasive capacity, and promotes a metastatic phenotype in osteosarcoma cell lines. Furthermore, by fixing canine primary osteosarcoma cells isolated in formalin and paraffin, they found a higher expression of miR-9 in osteosarcoma tumour cells when compared to healthy dog bones and osteoblasts. This suggests that miR-9 overexpression within tumour cells is primarily due to osteosarcoma tumour cells and not the inflammatory cells, often present in the tumour microenvironment. Furthermore, miR-9 is capable to enhance the sensitivity to ionizing radiation by suppression of NFkappaB1, which can affect the results of treatment in different types of tumours [54,55]. These findings reflect the possibility of using miR-9 as tumour biomarkers in human and canine osteosarcoma [11].

### 4.3. miR-196a

Aberrant expression of miR-196a is a frequent finding in several types of tumours and can induce both oncogenesis and tumour suppression. With the analysis of samples of appendicular tumours, previously diagnosed as high-grade tumours; Pazzaglia et al. (2015) demonstrated a decrease in the expression of miR-196a in osteosarcoma cell lines, and an elevated amount of its target protein Annexin 1, compared to normal tissue from the same species [56]. Annexin 1 has been shown to promote metastasis formation in breast cancer cells and facilitate cell migration and invasion. It also appears to play an important role in cell adhesion and migration routes. In fact, it is a mediator of vascular growth factor 1 (VEGF1)–induced cell migration. This study suggests that miR-196a could be involved in the progression of osteosarcoma in humans and dogs [56].

However, canine and human OSA cells display different responses to miR-196a overexpression, indicating that their expression profiles may depend on the species and the type of cell involved. Given the different results obtained on the expression and function of miR-196a and its targets in tumour progression, further studies are necessary to confirm that low levels of miR-196a in OSA cell lines can be used as prognostic biomarkers for disease progression [56].

### 4.4. MicroRNAs in the 14q32 Locus

Recent studies have shown that reduced levels of miRNA at the 14q32 locus are associated with an increased risk of metastasis and a worse survival rate in human patients suffering from osteosarcoma when compared to healthy bone tissues and normal osteoblasts [3,11]. Using quantitative PCR (qRT-PCR) to measure both miR-382 and miR-154 (representative transcripts of the 14q32 locus in humans), it was found that patients with the lowest levels of miR-382 had the highest risk of metastasis. These results show the prognostic utility that miR-382 could have in predicting the probability of developing metastatic tumours and the course of the disease in humans [3].

Another study showed that decreases in 14q32 miRNA levels stabilize the expression of the cMYC proto-oncogene in osteosarcomas, leading to the increased expression of oncogenic miR-17–92 [3,57]. Elevated levels of miR-17–92 groups, leads to an aberrant pattern of cell division and the inhibition of apoptosis in human osteosarcoma. Likewise, it was found that the restoration of these miRNAs (including miR-544, -369-3p, -134, 382) at the 14q32 locus, or a reduction of miR-17–92 groups by ectopic means, promotes osteosarcoma cellular apoptosis. These results suggest that decreases in miRNA at the 14q32 locus prevent apoptotic processes, and maintain tumorigenesis in osteosarcoma [57].

The previously cited findings, observed in human osteosarcomas, appear to be consistent with cross-species comparative analysis. Different studies have compared samples of osteosarcoma in reactive osteoblasts in dogs, finding a lower expression of miR-134 and miR-544 (orthologs to the cluster of human miRNA 14q32) in tumour cells compared [3,11]. Furthermore, reduced expression of miR-134 and miR-544 at the 14q32 locus in human osteosarcoma, as well as in its canine orthologs, are associated with shorter survival times. Therefore, this could suggest that deregulated miRNA groups at the 14q32 locus represent a conserved mechanism that contributes to the aggressive biological behaviour of osteosarcomas in the two species. Furthermore, these results support the idea that miRNAs at the 14q32 locus could be used as prognostic biomarkers, associating a low expression of the latter with osteosarcomas with an unfavourable prognosis [3].

### 4.5. miR-34a

Another miRNA notably associated with cancer is the miR-34 family. Composed of three members of the same family [58], these miR-34 have been shown to induce apoptosis and induce cell cycle arrest in the G1-G2 phase, in a p53-dependent manner. In addition their expression is decreased in osteosarcoma cell tissue samples and cell lines, compared to normal osteoblasts [11]. This information is in agreement with data generated in samples of human osteosarcoma, which suggests that the loss of miR-34a is a common process of the disease. Importantly, the loss of miR-34a in OSA is associated with shorter disease-free survival times and an overexpression of its different target genes involved in tumour formation, such as MET, SIRT1, and CDK6 [14,49]. Similarly, the low expression of miR-34a in human osteosarcomas has been shown to produce an increased expression of VEGF1, whose presence is related to the appearance of lung metastasis and, therefore, could be indicative of shorter survival times. These results highlight the potential of miR-34a as a biomarker, in which reduced levels of this miRNA would be indicative of a poor prognosis in canine OSA [14].

Whilst the most recent clinical studies carried out on miRNAs in cancer have concentrated on their use as potential prognostic and therapeutic biomarkers, their ability to target many messenger RNAs altered in tumours, makes these non-coding RNAs exceptional therapeutic targets to fight osteosarcoma [49]. In fact, Lopez et al. (2018) observed that stable overexpression of miR-34a in canine OSA cell lines reduced VEGF1 expression, with a concomitant decrease in cellular invasion and migration [14]. Furthermore, it significantly altered the transcriptional profile of OSA cells, including down-regulation of several genes involved in promoting the metastatic phenotype. This makes miR-34a a potential target in the search to produce a new generation of therapies for this disease [14]. Different studies have shown that the reintroduction of ectopic miR-34a through various methods, such as simulating miR-34a expression with chemically modified oligonucleotides or the design of miR-34a pro-drugs (tRNA/miR-34a) in canine osteosarcoma cell lines, reduces tumour cell viability, oncogenic growth, and cell migration and invasion, whilst increasing tumour cell apoptosis [14,49]. Furthermore, administration of the genetically engineered miR-34a pro-drug into a tumour in an experimental mouse model with orthotopic OSA xenografts resulted in decreased tumour growth, associated with increased tissue necrosis, as well as decreased lung metastasis. At the same time, it caused an increase in the sensitivity of malignant cells, to chemotherapeutic drugs. With the significant reduction of oncogenic growth, Alegre et al. (2018) were able to confirm the long-term efficacy of tRNA/miR-34a treatment on cell proliferation [49]. Taken together, this information exposes the clinical potential of these new miRNA-based therapies to fight osteosarcoma in dogs, and possibly also in humans.

### 4.6. Cluster of miR-106b-25 (miR-106b, miR-25 and miR-93-5p)

Cluster miR-106b-25 is composed of three different miRNAs, which are miR106b, miR25 and miR-93-5p and overexpression of miR-106b-25 cluster has been seen in most types of human cancers, such as gastric, hepatocellular, prostate, lung, and breast cancer and is related to poorer prognostics [59,60,61,62,63,64,65]. Likewise, an overexpression of miR-93-5p (a miRNA in the same cluster as miR-106b-25) has been reported in human OSA tissues [66,67]. Recently, a study carried out by Leonardi et al. (2020) also showed an overexpression of miR-93-5p in canine OSA [68]. In the same study, the expression of miR-106b and miR-25 were evaluated in dogs, finding no significant differences in their expression profiles when comparing healthy tissues with OSA tissues. Furthermore, they observed a higher expression of miR-93-5p in canine OSA cells compared to humans.

In a previous study, Kan et al. (2009) showed that miR-93 and miR106b have proliferative, antiapoptotic, cell cycle promoter and tumorigenic activities in vivo, and that their presence causes a decrease in p21 as well as its mRNA [61]. Studies have shown that the p21 protein is related to increased cell proliferation and the inhibition of apoptosis in different carcinomas, such as nasopharyngeal and hepatocellular carcinoma [69,70,71]. However, the expression of p21 is not affected in canine osteosarcomas which could be explained by an inhibitory activity in the progression of the cell cycle which is greater in OSA than in carcinomas [70]. Nevertheless, when a negative regulator of miR-93-5p was incorporated in OSA cells, the expression of p21 as well as its mRNA increased. This suggests that the control of miR-93-5p on cell proliferation and OSA progression could be mediated by different signalling pathways [69,72,73,74].

## 5. Circulating MicroRNAs and Their Potential as Non-Invasive Biomarkers

In dogs with appendicular osteosarcoma, even after amputation of the affected limb and after treatment with conventional chemotherapy, the prognosis remains uncertain. The use of new biomarkers could help distinguish between groups of patients with good and bad prognosis, allowing the latter to benefit from alternative, and possibly more aggressive therapeutic protocols in order to promote their recovery [20]. Unlike in human oncology, where several non-invasive blood-based biomarkers have been successfully identified, clinically available, non-invasive biomarkers have not yet been identified for canine tumours. Until now, some proteins and enzymes obtained in blood, such as canine C-reactive protein and serum ALP (alkaline phosphatase) levels, have been studied as potential low-invasive biomarkers in canine tumours. However, these protein biomarkers lack precision, since their activity can be altered by a multitude of causes that are not related to the cancer, such as liver damage, endogenous or exogenous use of corticosteroids, bone damage and inflammatory processes [20,75]. Consequently, new blood-based biomarkers could be very beneficial to allow a more accurate prediction of the course of the disease in patients with osteosarcoma [20]. Cancer cells not only show aberrant expressions of miRNA in affected tissues, but also appear to release these miRNAs into the bloodstream [75]. These “circulating miRNAs” are stable molecules and can be easily measured by reverse transcription techniques and subsequent quantitative PCR (RT-qPCR) [20]. With these techniques, it is possible to detected miRNAs concentrations around 10^−11^ M, while Tavaille et al. (2018) [76] have developed a specific detection method that was able to detect up to 10^−18^ M. Using commercial specific detection techniques to a single miRNA, between 8910 copies/µL plasma to 133,970 copies/µl can be detected, depending to miRNA examined [77] Furthermore, it has been observed that the circulating levels of miRNA in blood are able to accurately reflect the number of tumour cells present, to predict the response of patients to different treatments, to identify different clinical stages and to define the tumour grade. This shows the very interesting potential of miRNAs as possible prognostic and/or diagnostic biomarkers in neoplastic diseases in both humans and dogs [75].

In veterinary medicine, miR-214 and miR-126 were initially identified as deregulated miRNAs in cases of canine hemangiosarcoma [20]. These two miRNAs are involved in the regulation of angiogenesis, proliferation, migration, and cell death of cancer cells and, therefore, their inefficient regulation has a considerable influence on tumour progression and development [20]. Furthermore, these studies demonstrated that there are no significant correlations between the expression of miR-214 and miR-126 with other clinical parameters, such as age, weight, or the presence of any concurrent disorders [20,50,51,75].

In a study by Heishima et al. (2017), it has been shown that there is a significant increase in circulating miR-214 and miR-126 levels in a wide spectrum of canine cancers, and that they are associated with short survival times [75]. While miR-214 overexpression showed special prognostic potential in sarcomas, miR-126 levels were elevated in a broader range of epithelial and non-epithelial tumours, not including chondrosarcomas and lymphomas. It should be noted that in this study, they had to exclude osteosarcomas because they exhibited extraordinarily high levels of circulating miR-214, suggesting that osteosarcomas have a different profile of circulating miRNA compared to other non-epithelial tumours [75].

In a more recent study, it was found that circulating levels of miR-214 and -126, prior to treating dogs with appendicular OSA, could predict the time to metastasis and the time to death after amputation and chemotherapy treatment. In this study, high levels of miR-214 predicted shorter survival times while high levels of circulating miR-126 were associated with longer survival times [20]. In humans with osteosarcoma, miR-214 promotes the growth, invasion, metastatic behaviour, survival, and chemoresistance of tumour cells. In contrast, high levels of miR-126 prevent proliferation, migration, invasion, epithelial-mesenchymal transition, and chemoresistance of osteosarcoma cells. This study highlights the similarities between human canine species regarding the regulation of miR-214 and miR-126 and suggests that these miRNAs have potential as predictive biomarkers in OSA [20].

## 6. Conclusions

OSA are a rare form of cancer with high death rates in humans and dogs. The discovery of miRNAs marked a new era in molecular biology and their presence can give valuable information about the physiopathology of osteosarcomas as well as other carcinomas in humans and dogs. However, much of the evidence for the use of miRNAs as biomarkers in dogs is derived from studies in cell lines and do not have any correlation to clinical parameters. Thus, the use of miRNAs as biomarkers in canine OSA indeed holds promise, but further studies are necessary for this to become a reality in veterinary clinical practice. Only the study on miR-214 and 126 actually correlate a clinical parameter with a difference in miRNAs levels. Molecular techniques to detection of miRNAs are increasingly economically feasible for veterinary clinical practice and different studies are being carried out in order to detect possible miRNAs to facilitate, not only prognostic and prediction, but also for future therapeutic targets. In future, the aberrant expression of miRNAs associated with tumour progression could be useful in serving as prognostic and predictive biomarkers, or early detection of primary or metastases.

## Figures and Tables

**Figure 1 vetsci-07-00146-f001:**
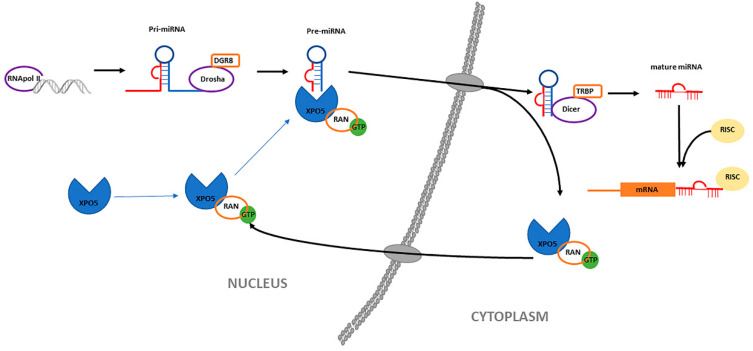
Diagram of miRNA biogenesis. In the nucleus, miRNA that are transcribed by RNA polymerase II or III as primary transcripts (pri-miRNA). The enzyme Drosha and its cofactor DGCR8/Pasha then cut the pri-miRNA molecule to form a pre-miRNA. This pre-miRNA is actively transported from the nucleus to the cytoplasm by the nuclear transport receptor Exportin 5 (XPO5) in a Ran-GTP protein dependent manner by XPO5. In the cytoplasm, the pre-miRNA is cut by a second enzyme called Dicer, until the formation of a mature and short double-stranded miRNA molecule. The miRNA duplex unwinds, and is incorporated into the RISC protein complex.

**Table 1 vetsci-07-00146-t001:** miRNAs related to OSA, target of these miRNAs, result of aberrant expression and possible uses.

miRNAs	Expression Profiles in OSA	Target Genes or Proteins	Result of Aberrant miRNA Expression	Possible Uses
miR-1 and miR-133b	Reduced expression	MET and MCL1	Cell proliferation and growth Cancer cells Invasion Cancer cells Survival Inhibition of apoptosis	Diagnostic biomarker for canine OSA
miR-9	Overexpression	Gelsolin	Cell motility and invasive capacity increase Promotion of metastasis	Possible prognostic biomarker
miR-196a	Reduced expression	Annexin 1	Facilitates migration and invasion Cancer cells adhesion and migration Mediates VEGF1 induced cell migration	Potential prognostic biomarker
14q32 locus (miR-544, miR-369-3p, miR-134 and miR-382)	Reduced levels	cMYC	Promotion of metastasis Poorer survival rates Inhibition of apoptosis Tumorigenesis maintenance	Potential prognostic biomarker Potential therapeutic target
miR-34a	Reduced levels	MET, SIRT1, CDK6, VEGF1	Mediates VEGF1 induced cell migration Shorter disease-free survival times	Potential prognostic biomarker Potential therapeutic target
miR-106b-25 cluster (miR-106b, miR-25 and miR-93-5p)	Overexpression	p21	Cell proliferation and growth Cancer cells Invasion Cell cycle promotors Activation of tumorigenesis	Potential prognostic biomarker
Circulating miR-214	Increased levels in circulation		Shorter survival times	Potential non-invasive predictive biomarker
Circulating miR-126	Increased levels in circulation		Longer survival times	Potential non-invasive predictive biomarker

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
