# Peer review of "MicroRNAs as Biomarkers in Canine Osteosarcoma: A New Future?"

_vetsci, 2020, doi:10.3390/vetsci7040146_

Round 1
Reviewer 1 Report
Overall, this paper is very interesting and illustrates the benefits of a one health approach to studying molecular mechanisms driving osteosarcoma.
Specific comments:
1) A brief sentence to define miRNAs are should be included in the abstract;
2) Including a diagram of the biogenesis of miRNA production would aid understanding;
3) A discussion of how conserved miRNA are between species should be included;
4) What concentration range of miRNAs are detectable in patient blood? How does this relate to the sensitivity of the assay used to detect miRNAs?;
5) Are any miRNA targeted treatments in clinical trials yet? What are the advantages and disadvantages? Similarly, are any validated miRNA biomarkers routinely used in clinical practice?; and
6) The conclusion should be expanded to include more blue sky thinking regarding the details of how analysis of miRNAs could be included into clinical practice, and does this differ between human and veterinary medicine given that in humans the goal is to cure the disease whereas in dogs we are often more concerned with maintaining quality of life.
Reviewer 2 Report
The review is interesting. The only issue is the English: a mix of UK-USA style. More USA than English, so I suggest at least to use “tumor/s” instead of the English “tumour/s”.
Here below some suggestions:
Lines 12-13: change with “ Sarcomas are frequent in dogs and canine species is an excellent animal model for studying the human counterpart”
Lines 14-16: change with “miRNAs could give a contribute in the diagnosis and prognosis of different types of tumours in animal species, as already in humans”
Line 40: if Authors write “more” it is necessary to compare. Maybe better “Compared with other species OSA occurs more frequently in dog”
Line 63: delete “has”
Line 66: “and therefore they are”
Line 69: “carrying out”
Line 152: “Recent studies have shown” because references at the end of the sentence are two.
Line 167: “observed” instead of “founded”
Line 185: “low” instead of “under”
Line 193: “observed” instead of “found”
Line 253: are involved in the regulation of – instead of “have the ability to regulate”
Line 261: short instead of ”worse”
Line 274: similarities that dogs and humans have – change wit “ similarities between human and canine species regarding….”
Line 276: change “this disease” with “OSA”
Line 278: Sarcomas are a rare form of cancer. This is not true. Sarcoma, in canine species, can’t be defined as rare. Please change sarcoma with “OSA”.
Line 286: delete “future”
Reviewer 3 Report
Dear Author,
In my opinion, the article is a well-prepared manuscript with all necessary parts: abstract, introduction, and explanation. All arrangements are well described. References are very actual from the last 20 years.
Dog and their tumors are a perfect model for the study on human tumors, for example, lymphoma, mammary carcinoma, and featured osteosarcoma. The new diagnostics method, like using miRNAs or a survey of “liquid biopsy,” is auspicious in human and veterinary medicine.
The use of these methods will help in quick diagnosis, facilitate therapy, and extend the survival time.
The last sentence from the abstract is very important: “However, further studies are needed to make the use of miRNAs as biomarkers a common practice.”
Best regards